# The Experience of a Single Tertiary Center Regarding Benign and Malignant Tumors in Acromegalic Patients

**DOI:** 10.3390/medicina59061148

**Published:** 2023-06-15

**Authors:** Iulia-Stefania Plotuna, Melania Balas, Ioana Golu, Daniela Amzar, Adrian Vlad, Lavinia Cristina Moleriu, Mihaela Vlad

**Affiliations:** 12nd Department of Internal Medicine—Discipline of Endocrinology, “Victor Babes” University of Medicine and Pharmacy Timisoara, P-Ta Eftimie Murgu 2, 300041 Timisoara, Romania; iulia.plotuna@umft.ro (I.-S.P.); balasm12@yahoo.com (M.B.); igolu25@yahoo.com (I.G.); dana_amzar@yahoo.com (D.A.); vlad.mihaela@umft.ro (M.V.); 2Department of Endocrinology, County Emergency Hospital Timisoara, Blvd. Liviu Rebreanu 156, 300723 Timisoara, Romania; 3Center for Molecular Research in Nephrology and Vascular Disease, “Victor Babes” University of Medicine and Pharmacy Timisoara, P-Ta Eftimie Murgu 2, 300041 Timisoara, Romania; 42nd Department of Internal Medicine—Discipline of Diabetes, Nutrition and Metabolic Diseases, “Victor Babes” University of Medicine and Pharmacy Timisoara, P-Ta Eftimie Murgu 2, 300041 Timisoara, Romania; 5Department of Functional Sciences—Discipline of Medical Informatics and Biostatistics, “Victor Babes” University of Medicine and Pharmacy Timisoara, P-Ta Eftimie Murgu 2, 300041 Timisoara, Romania; moleriu.lavinia@umft.ro

**Keywords:** acromegaly, tumors, thyroid, diabetes mellitus, obesity

## Abstract

*Background and Objectives:* Acromegaly is a rare disease associated with increased levels of growth hormones (GHs) that stimulates the hepatic production of insulin growth factor-1 (IGF-1). Increased secretion of both GH and IGF-1 activates pathways, such as Janus kinase 2/signal transducer and activator of transcription 5 (JAK2/STAT5), and mitogen-activated protein kinase (MAPK), involved in the development of tumors. *Materials and Methods:* Given the disputed nature of the topic, we decided to study the prevalence of benign and malignant tumors in our cohort of acromegalic patients. In addition, we aimed to identify risk factors or laboratory parameters associated with the occurrence of tumors in these patients. *Results:* The study group included 34 patients (9 men (25.7%) and 25 women (74.3%)). No clear relationship between the levels of IGF-1 or GH and tumor development could be demonstrated, but certain risk factors, such as diabetes mellitus (DM) and obesity, were more frequent in patients with tumors. In total, 34 benign tumoral proliferations were identified, the most common being multinodular goiter. Malignant tumors were present only in women (14.70%) and the most frequent type was thyroid carcinoma. *Conclusions:* DM and obesity might be associated with tumoral proliferation in patients with acromegaly, and findings also present in the general population. In our study we did not find a direct link between acromegaly and tumoral proliferations.

## 1. Introduction

Acromegaly is a rare chronic disease associated, in most cases, with increased levels of GH from a benign pituitary adenoma [1]. It has an insidious evolution; as a result, the diagnosis is usually established years after its onset [2]. In most of the patients, acromegaly coexists with several comorbidities, such as cardiovascular diseases, DM, or sleep apnea [1].

Due to novel therapies, the life expectancy of patients suffering from this pathology has increased [3]. This might explain the surge in tumoral proliferation in this population [4]. The most frequent malignancies associated with acromegaly are colorectal, thyroid, and breast cancers [5].

The pathophysiology of tumors is a multistep process involving many elements [6]. Cancer development starts with the aberrant multiplication of a single cell. Usually, this process unfolds gradually, and several other alterations emerge along the way, explaining why this pathology is usually diagnosed later in life [6].

There are many factors that may increase the risk of developing cancer, like radiation, smoking, or alcohol [7]. In addition, chronic diseases, such as obesity, DM, and sleep apnea, are supposed to be involved in promoting cancer through multiple mechanisms [8]. In fact, the diseases mentioned above are present in many acromegalic patients [9]. Growth hormone/insulin growth factor-1 (GH/IGF-1) axis has been associated with cancer risk due to its involvement in cell multiplication through mitogen-activated protein kinase (MAPK) signaling [10], and it also enables the mammalian target of rapamycin pathway (mTOR), which is involved in the genesis of tumors in tissues from obese and/or diabetic mice [8]. Furthermore its main pathway for gene transcription is JAK-2/STAT, which is involved in the pathogenesis of cancer growth [10]. Another important factor that favors the development of tumors, such as, breast, colon, prostate, and thyroid, is the presence of an increased number of insulin receptors, which are caused by IGF-1 [11].

The current data about benign tumors in acromegalic patients are scarce, but some studies show an increased risk of goiter, colon polyps, and skin tumors [12]. Nevertheless, it is mentioned that these findings might be a result of better medical screening [5,13].

Our aim was to investigate the prevalence of benign and malignant proliferations in our group of acromegalic patients. We also wanted to identify if particular risk factors or laboratory parameters might be related to tumoral development in acromegaly.

## 2. Materials and Methods

### 2.1. Study Design

We conducted a case–control study, in accordance with the Declaration of Helsinki. This study was approved by the local Ethics Committee (approval number 88/2020). The study group was represented by a cohort of patients diagnosed with acromegaly between 2001 and 2022 in the Department of Endocrinology of the County Emergency Hospital in Timisoara, Romania. The collected data included personal history, family history of cancer, physical examination, laboratory assessment, magnetic resonance imaging (MRI), and the prescribed treatment. All the patients underwent neck ultrasonography (US) for thyroid pathology.

The inclusion criteria for this study were age at diagnosis > 18 years, clinical signs and symptoms of acromegaly certified by GH > 0.4 ng/mL during oral glucose tolerance test (OGTT), and the increased level of IGF-1 for age and gender. In patients with DM, OGTT could not be performed, so the mean value of 4 GH determinations over 24 h was used instead (a mean value > 1 ng/mL confirmed the diagnosis of acromegaly). The exclusion criteria were the absence of medical records confirming the diagnosis of acromegaly and the lack of medical follow-up after the diagnosis.

Controlled disease was defined by the IGF-1 level, determined by a reliable standardized assay through chemiluminescence, situated in the age-adjusted normal range, and by the GH level, determined from a random sample by a reliable standardized assay through chemiluminescence (lower than 1.0 ng/mL). Partial disease control was defined as follows: a GH level, determined from a random sample by a reliable standardized assay through chemiluminescence, higher than 1.0 ng/mL but reduced by ≥50% in comparison to its baseline value, or by the IGF-1 value >1.3 × upper limit of normal (ULN) but reduced by ≥50% in comparison to its value before therapy. The disease was uncontrolled if one of the following criteria was met: symptoms specific for acromegaly; basal serum GH > 1 ng/mL (random sample) that did not decrease by ≥50% compared to the basal value; IGF-1 >1.3 × ULN that did not decrease by ≥50% from baseline; and tumor progression.

The disease duration was calculated from the occurrence of the first symptom of acromegaly or the onset of the first complication. This information was retrieved either from the patient’s medical records (available in most cases) or from the family history.

The patients were screened annually for complications according to the national guidelines for the diagnosis and management of acromegaly [14]. The date of occurrence and the management of each complication were recorded. Information regarding the patient’s benign and malignant tumors was retrieved from the hospital’s database and the patients’ medical files (previous hospital admissions or imaging tests). Thyroid nodules were stratified according to the American College of Radiology Thyroid Imaging, Reporting and Data System (ACR TI-RADS) [15]. Recommendations for fine-needle aspiration or follow-up were based on a nodule ACR TI-RADS score. The tumors were classified as malignant if a pathological report confirmed the diagnosis. Mammography was recommended if the patients had a palpable breast nodule or a family member diagnosed with breast cancer. US and prostatic exams were performed in all male subjects. Polysomnographic evaluation was performed on all patients.

### 2.2. Statistical Analysis

The statistical analysis was performed using Microsoft Excel and JASPv16.4 programs. Continuous variables are presented as mean ± standard deviation (SD or median (range), and categorical variables as numbers (%). The following statistical tests were used: Shapiro–Wilk for the normality of data distribution, and Mann–Whitney, Kruskal–Wallis and chi-square (χ2) for the significance of differences. The odds ratio (OR) parameter was calculated, and the 95% confidence interval was estimated to assess if certain factors are associated with tumor development in acromegalic patients. The level of significance was set at *p* < 0.05 for the whole study.

## 3. Results

The study group included 34 patients: 9 men (25.7%) and 25 women (74.3%). Patients’ ages ranged between 20 and 64 years when acromegaly was diagnosed. Men were older than women, and they had a longer duration of undiagnosed disease. The age at diagnosis of the first tumor was similar in both genders (Table 1 and Table 2). Obesity did not seem to be gender specific, while DM was diagnosed exclusively in women (Table 1). There were some differences in the biological parameters between men and women, without reaching the threshold for significance (Table 2).

In most cases, the first-line treatment was surgery. Disease control was obtained through surgery in 25% of the patients. For the rest of them, medical therapy was administered, and 17.6% of the patients underwent radiotherapy.

In total, 34 benign tumoral proliferations were identified. Out of these, 17 (50%) were represented by multinodular goiter (Table 3). In total, 33 patients (67.6%) were diagnosed with different forms of tumors. Among these, 18 (52.9%) had only benign proliferation and 5 (14.7%) had at least one malignant tumor. Eight patients were diagnosed with two tumors (23.5%), two patients with three tumors (5.9%), and one patient with four tumors (2.9%). As well, the most frequent first tumor (41.2%) was multinodular goiter. Four patients (11.8%) diagnosed with malignant tumors had at least one other benign proliferation.

The most frequent benign proliferation among female patients was multinodular goiter (60%, Table 4), while the most frequent benign pathology among men was benign prostatic hypertrophy (33.3%, Table 5). Only women had malignancies (14.70%), the most common being papillary thyroid carcinoma (three cases), which did not have any specific histologic pattern of aggressiveness (Table 4).

The analysis of the temporal relationship between the diagnosis of the tumor and that of acromegaly showed the following distribution: in 4 patients (11.8%), the tumor preceded acromegaly, and in 13 patients (38.2%), both diseases were diagnosed in the same year, while 6 patients (17.6%), were diagnosed with tumors after acromegaly.

There was no significant correlation between factors concerning acromegaly, such as age at diagnosis, IGF-1 value, IGF-1 ULN, disease duration, GH, level of sexual hormones, or the risk of developing malignant tumors. Disease control is the primary therapeutic target in every patient, and it is associated with a decrease in morbidity and mortality. In our group, neoplasms were diagnosed more frequently in patients who had partial or uncontrolled disease (Table 6). The patients were questioned about their personal or family medical history, and two female patients mentioned thyroid pathology, while one declared the diagnosis of a breast tumor in a first-degree relative.

The existence of a correlation between known risk factors for cancer, such as obesity, DM, and sleep apnea, was searched for. DM (*p* = 0.037, OR = 1.6, OR ∈ [1.2;2.2]) and obesity (*p* = 0.019, OR = 7, OR ∈ [1.2;39.6]) are major risk factors for developing tumors (Table 7). One other important risk factor associated with the presence of a tumor, in our study-group, was female gender (*p* = 0.045, OR = 5, OR ∈ [1.0;25.8]) (Table 6).

## 4. Discussion

Given the uneven cancer distribution between genders and the plethora of risk factors associated with acromegaly, we searched for significant clinical and biological differences between women and men in our study. Furthermore, there is scientific evidence for estrogens that can stimulate GH secretion by decreasing IGF-1 production [16], while testosterone enhances GH and, therefore, IGF-1 production [17]. Additionally, there are studies published that underline the contribution of GH and IGF-1 in the process of tumor formation by stimulating cell proliferation, leading to increased mitosis [18].

Our study found no relevant differences between men and women regarding the age at diagnosis, GH and IGF-1 levels, or duration of the disease. These findings could be influenced by the small number of male patients included in our study.

This finding was also mentioned in a large survey, carried out on 50,170,946 subjects that reported a female incidence of 12 cases per million person per year and a male incidence of 10 cases per million person per year [19]. Dal J. et al. also mentioned a larger female population, but the gender distribution over the years became even [20]. The Mean age at diagnosis, nadir GH, and pituitary adenoma size were similar between women and men, but the values of IGF-1 were lower in women, aspects encountered in our study as well.

We found that women were more prone to develop tumors, an aspect also mentioned in a large Italian study, that associated this finding with the fact that women were older than men [21].

Possible correlations between DM and neoplasia were mentioned as early as 1932, and cancers associated with this pathology were in the pancreas, liver, endometrium, breast, colon, and urinary bladder [22]. Cancer cells have increased energy needs, and the presence of a high glycemic load in patients with DM seems to facilitate tumoral proliferation and growth [23]. One of the biological mechanisms involved mentions insulin resistance and IGF-1 [24]. Insulin reduces the circulating levels of the IGF binding proteins (IGFBP1 and IGFBP2), which leads to increased levels of circulating IGF-1 [24]. Consequently, the receptors for insulin and IGF-1 are stimulated, and this induces the activation of the mTOR pathway, which is the common denominator in multiple tumors and normal tissues of obese and diabetic mouse models [8]. Moreover, insulin and IGF-1 are also promoting mechanisms that are involved in tumoral progressions, such as proliferation and angiogenesis [25]. Another factor that increases the risk for both DM and cancer is obesity, which is associated with a higher probability of developing cancer in multiple anatomic sites, such as colorectal, postmenopausal breast, ovaries, urinary bladder, and thyroid [26]. Multiple metabolic pathways, along with inflammatory and immunologic alterations associated with fat deposition, affect deoxyribonucleic acid (DNA) repair, gene function, as well as epigenetic changes, and permit malignant transformation and progression [27]. There are two large studies that demonstrated that DM is a risk factor associated with tumors in patients with acromegaly, such as our findings [21,28].

A higher number of female patients diagnosed with DM was also observed by Monteros et al. in a study performed on 257 acromegalic patients [29]. This was associated with older age, macroadenoma, disease duration, and a basal GH > 30 μg/dL [28]. These aspects were not significantly different between the genders in our study. Moreover, our male patients had a longer disease duration (11.9 ± 5.4 years vs. 9.3 ± 6.2 years) and were older at diagnosis (50.0 ± 5.4 years vs. 45.0 ± 11.4 years) in comparison to women.

A large Mexican study also observed an increased prevalence of DM in women, but with a mean GH value higher in men [30], an aspect not present in our population of acromegalic patients (. The general prevalence of DM is higher in men, but this aspect changes after the age of menopause when metabolic alterations lead to the impairment of insulin secretion and insulin sensitivity [31]. In our study, nine women (36 %) were diagnosed with secondary hypogonadism, because of tumor compression, which led to a decrease in estrogen levels and, possibly, to changes similar to that of menopause. The body mass index did not differ significantly between groups, while DM was diagnosed only in women.

In Romania, the most frequent types of cancer in 2020, according to Globocan (an interactive web-based platform presenting global cancer statistics to inform cancer control and research), were colorectal (13.1%), lung (12.3%), breast (12.2%), prostate (8.1%), and urinary bladder (5.2%), while thyroid cancer represented only 1.7% of all the cases [32]. The most frequent type of malignancy in our patients was thyroid cancer, namely the papillary form. This type of malignancy is more frequent in women, being usually diagnosed between 40 and 60 years of age [33]. This aspect together with the higher number of female patients might explain our results. Current guidelines do not include acromegaly as a risk factor for developing thyroid cancer, so additional screening methods are not recommended for these patients [34].

Nevertheless, it has been suggested that thyroid cells harbor IGF-1 receptors that potentiate the proliferative effect mediated by thyroid-stimulating hormones (TSHs) [35]. In a study that included 14 patients with differentiated thyroid cancer (DTC), the most common genetic alteration was in the NRAS codon 61, whereas BRAF V600E was less frequent [35]. Conversely, a recent study concluded that the BRAF V600E mutation was present in 14.3% of the patients diagnosed with DTC and acromegaly, while none of them harbored the NRAS codon 61 mutation [36].

A large case–control study published in 2012, including 124 acromegalic patients (of whom 61.3% were women), discovered that 54% of the patients had thyroid nodules and nine had papillary thyroid carcinoma (PTC), suggesting an increased prevalence of thyroid cancer (*p* = 0.0011) [37]. Similar results were published in a cohort of 1512 acromegalics that was followed-up for 10 years, where the risk of thyroid cancer was increased (SIR 3.99; 95% CI, 2.32–6.87, *p* < 0.001), suggesting that the family history of cancer, long disease duration before diagnosis, the presence of DM, and previous pituitary radiotherapy are risk factors [21].

In both studies, women were prevalent, and they had about the same mean age as men (45 and 47 years, respectively), aspects that were present in our cohort as well. In the first study, 7.2% of the patients were diagnosed with thyroid cancer, most of them women (66%) [37]. The results are consistent with ours regarding female prevalence, the high number of thyroid cancers, and the systematic morphological evaluation of the thyroid.

Another retrospective study in which cancer screening was implemented, including 160 patients with acromegaly (mean age 52 years), found malignant tumors in 21.3% of them without identifying a certain risk factor [38]. Thyroid cancer was the most frequent (17.1%), followed by breast and colorectal cancer [38]. In our study, about the same proportion of the patients had malignant tumors (14%), but with a female preponderance, most likely due to their high number compared to men. A Korean study also supported the premise that women are more likely to develop malignancies and proposed delayed diagnosis and sustained exposure to GH as possible risk factors [39].

An important difference related to the abovementioned studies is the fact that we did not find any cases of malignancy in the breast or renal/urinary tract, most likely due to the small number of patients, especially men (who are more likely to develop renal malignancies) [21].

The association between colon polyps and colon cancer in acromegaly has been documented in multiple studies. It has been linked to the antiapoptotic effects of the GH/IGF-1 axis, described in some colonic adenocarcinomas, and to the expression of peroxisome proliferator-activated receptor (PPAR) gamma, a tumor suppressor gene [40]. These changes were observed mainly in patients with untreated acromegaly [40]. In some studies, colon polyps had a prevalence of up to 66.8% and were more frequent in the descending colon, without having distinctive pathological features [41].

While adenomatous polyps were present in three of our acromegalic patients, there was only one case of colorectal cancer (namely papillary adenocarcinoma, stage G2, pTa4N2M1), which was very aggressive and caused the death of our patient. This case had a disease duration of more than 10 years, but she was less than 50 years old at the diagnosis of the colon cancer and had no family history of malignancy. These findings confirm the necessity of colonoscopy in patients with acromegaly, at diagnosis, as the current guidelines recommend [34].

Benign tumors were found in most of our patients. They did not seem to influence morbidity and mortality in a significant way, but they are a cause of concern when it comes to quality of life and possible malignant transformation. It should be noted that even though they are benign, endometrial fibroadenomas in women are associated with infertility (2–3% of the cases) and anemia [42,43], while benign prostatic hypertrophy in men may cause sexual dysfunction, renal insufficiency, or urinary tract infections [44].

According to our findings and to some previously published data [12,37], we conclude that thyroid cancer is frequent in patients with acromegaly and suggest that US screening for thyroid nodules should be implemented according to the national guidelines. A factor that needs to be considered, though, is the heterogeneity of the population, with different risk factors for thyroid cancer and different access to therapy and surveillance. An increase in the burden of cancers is associated with their advanced stages [45,46], so an earlier diagnosis is recommended. It should also be noted that age was associated with disease-specific survival in patients with thyroid cancer [47]. Three of our five patients with malignancies were diagnosed after acromegaly was detected, possibly due to the screening performed.

### Limitations

Our results might be biased because we performed neck ultrasonography in all the patients, a maneuver mentioned as an explanation for the increase in the number of thyroid cancers.

Another limitation is the small number of patients from the cohort. However, some of our results are similar to those mentioned in larger studies.

## 5. Conclusions

Our data revealed that malignancies associated with acromegaly were detected only in women. The most common form is papillary thyroid carcinoma, which did not have any specific histologic pattern of aggressiveness. Several factors might increase the risk of tumors in acromegalic patients, such as female gender, DM and obesity. DM is diagnosed more frequently in women, with acromegaly possibly related to low levels of feminine sexual hormones and to age. The most frequent benign proliferation among women was multinodular goiter (60%), while the most frequent benign pathology among men was benign prostatic hypertrophy.

## Figures and Tables

**Table 1 medicina-59-01148-t001:** Clinical characteristics, disease control, and comorbidities of the patients diagnosed with acromegaly.

Characteristic	Women (*n* = 25)	Men (*n* = 9)	Total (*n* = 34)	*p*
BMI (kg/m^2^), mean ± SD	29.68 ± 6.99	31.4 ± 3.28	30.1 ± 6.2	0.261
Age at diagnosis (years), mean ± SD	45.0 ± 11.44	50.0 ± 5.41	43.0 ± 11.5	0.490
Age at the diagnosis of the first tumor (years), mean ± SD	46.42 ± 10.27	48.6 ± 7.82	35.82 ± 21.88	0.482
Disease control, *n* (%)	C = 13 (52)	C = 4 (44.4)	C = 17 (50)	0.305
	PC = 5 (20)	PC = 4 (44.4)	PC = 9 (26.47)	
	UC = 7 (28)	UC = 1 (11.2)	UC = 8 (23.52)	
DM, *n* (%)	8 (32)	0 (0)	8 (23.5)	0.075
Surgical therapy, *n* (%)	20 (80)	8 (88.9)	28 (82.4)	1
Gonadotropin deficiency, *n* (%) *	9 (36)	6 (66.7)	15(44.1)	0.1

**Legend:** DM = diabetes mellitus, C = controlled, PC = partial control, UC = uncontrolled, * = FSH and LH deficiency were recorded if they were a consequence of adenoma compression.

**Table 2 medicina-59-01148-t002:** Biochemical characteristics of the patients diagnosed with acromegaly.

Characteristic	Women (*n* = 25)	Men (*n* = 9)	Total (*n* = 34)	*p*
GH at diagnosis (ng/mL), mean ± SD	17.85 ± 20.4	13.80 ± 9.61	17.65 ± 22.23	0.573
IGF-1 at diagnosis (ng/mL), mean ± SD	625.2 ± 327.2	642.67 ± 151.57	629.8 ± 288.0	0.879
IGF-1 ULN, mean ± SD	2.62 ± 1.3	2.76 ± 0.96	2.73 ± 1.26	0.942
Estimated diagnostic delay (years), mean ± SD	9.32 ± 6.16	8.44 ± 5.61	10.35 ± 6.11	0.711
IGF-1 ULN × estimated diagnostic delay, mean ± SD	19.35 ± 18.98	25.63 ± 19.96	21.0 ± 19.1	0.407
IGF-1 at diagnosis × estimated diagnostic delay, mean ± SD	4282.1 ± 3801.2	5714.3 ± 3851.0	46,661.3 ± 3809.9	0.341

**Legend:** GH = growth hormone, IGF-1 = insulin growth factor-1, IGF-1 ULN = ratio between IGF-1 level and upper limit of the normal range adjusted for age and gender.

**Table 3 medicina-59-01148-t003:** Number of benign and malignant tumors in patients with acromegaly.

Localization	Benign (*n*)	Malignant (*n*)
Thyroid	17	3
Uterus	5	0
Prostate	3	0
Colon	4	1
Parathyroid	2	0
Ovary	1	0
Lungs	1	0
Skin	0	1
Pancreas	1	0
Total	34	5

**Table 4 medicina-59-01148-t004:** Neoplasm distribution in women.

Women	Benign (*n*)	Malignant (*n*)
Thyroid	15	3
Uterus	5	0
Large intestine	3	1
Parathyroid	2	0
Ovary	1	0
Lungs	1	0
Skin	0	1
Pancreas	1	0
Total	27	5

**Table 5 medicina-59-01148-t005:** Neoplasm distribution in men.

Men	Benign (*n*)	Malignant (*n*)
Prostate	3	0
Thyroid	2	0
Colon	1	0
Total	6	0

**Table 6 medicina-59-01148-t006:** Clinical and biochemical parameters in patients with tumors and acromegaly.

Parameter	Patient with Tumors (*n* = 23)	Patients without Tumors (*n* = 11)	*p*
Men/Women, *n*	4/19	5/6	0.045
BMI (kg/m^2^), mean ± SD	30.06 ± 4.82	30.59 ± 8.72	0.837
Age at acromegaly diagnosis (years), mean ± SD	45.0 ± 10.64	41.0 ± 12.78	0.308
Estimated diagnostic delay (years), mean ± SD	8.65 ± 6.58	5.5 ± 3.8	0.155
Disease duration (years), mean ± SD	8.65 ± 6.54	10.45 ± 4.08	0.412
GH at diagnosis (ng/mL), mean ± SD	18.55 ± 20.92	22.56 ± 32.53	0.665
IGF-1 at diagnosis (ng/mL), mean ± SD	666.6 ± 325.6	552.8 ± 180.6	0.200
IGF-1 ULN, mean ± SD	2.92 ± 1.34	2.30 ± 0.95	0.185
IGF-1 ULN × estimated diagnostic delay, mean ± SD	23.46 ± 19.52	14.60 ± 16.39	0.203
IGF-1 at diagnosis × estimated diagnostic delay, mean ± SD	5158.8 ± 3804.3	3427.1 ± 3432.4	0.217
Disease control, *n* (%)	C =10 (43.5)	C = 7 (63.6)	0.358
	PC = 6 (26.1)	PC = 3 (27.3)	
	UC = 7 (30.4)	UC = 1 (9.1)	

**Legend:** BMI = body mass index, GH = growth hormone, IGF-1 = insulin growth factor-1, IGF-1 ULN = ratio between IGF-1 level and upper limit of the normal range adjusted for age and gender, C = controlled, PC = partial control, UC = uncontrolled.

**Table 7 medicina-59-01148-t007:** Comorbidities in patients with acromegaly and tumors.

Parameter	Patient with Tumors (*n* = 23)	Patients without Tumors (*n* = 11)	*p*
Obesity, *n* (%)	20 (87.0)	6 (54.5)	0.019
Diabetes mellitus, *n*, (%)	8 (34.8)	0 (0)	0.037
Gonadotropin deficiency, *n* (%) *	13 (56.5)	5 (45.5)	0.712
Sleep apnea, *n* (%)	Present 12 (52.2) Absent 11 (47.8)	Present 6 (54.5) Absent 5 (45.5)	1

**Legend:** * = FSH and LH deficiency were recorded if they were a consequence of adenoma compression.

## Data Availability

The data presented in this study are available on request from the corresponding author.

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
