# Peer review of "The Experience of a Single Tertiary Center Regarding Benign and Malignant Tumors in Acromegalic Patients"

_medicina, 2023, doi:10.3390/medicina59061148_

Round 1
Reviewer 1 Report
The manuscript "The experience of a single tertiary center regarding benign and 3 malignant tumors in acromegalic patients" by Plotuna et al. reports a single-center analysis of the occurrence of tumors in a group of 34 patients with acromegaly, especially looking at predisposing risk factors in that particular patient group.
GENERAL REMARKS
The manuscript is generally well-composed and presents an interesting perspective within the field, even though it is not very novel. It is mostly well written and readable, but the wording is odd at times and imprecise.
Regarding the usage of abbreviations, it would enhance the readability of the manuscript if each term is first written out in full with the abbreviation provided in parentheses afterwards. For example, instead of using "DM (diabetes mellitus)", the authors should consider using "diabetes mellitus (DM)". This format is more conventional and reader-friendly, especially for those who might not be intimately familiar with the specific abbreviation.
ABSTRACT
The abstract summarizes the study well, but just as the whole manuscript needs to be overhauled. "Malignant tumors" are mentioned, but no frequency is reported. The conclusions report that diabetes mellitus and obesity are risk factors for cancer in patients which is true for the general population as well. So maybe the conclusion should read along the lines of "Just as in the general population diabetes mellitus and obesity are associated with tumors in patients with acromegaly, but not specific features of acromegaly." Please avoid the term "cancer" when also referring to benign lesions such as [inconspicuous] thyroid nodules.
further comments:
* line 21: "seems to activate certain pathways" - please reword, it's very imprecise
* line 22: "Given the incomplete data and controversy of the subject" - please reword
* line 24: "certain parameters" - please reword
* line 31: last word is "obesity" but should read "acromegaly" (?)
INTRODUCTION
The introduction explains basic epidemiological data of acromegaly and considerations on the reasons for developing neoplasms in this particular disease. While briefly presenting some of these aspects, the introductions remains brief and rather superficial. Details are not given. Furthermore, the introduction is not concluding with statements why the present work was conducted and what research questions it aims at addressing, which would have been a helpful addition.
further comments:
* line 36/37: "Acromegaly is a rare chronic disease (2.8 up to 13.7 cases per 100,000 population) associated, in most cases, with increased levels of GH" - without the parentheses it essentially reads "Acromegaly is a rare chronic disease associated with increased levels of GH"
* line 55: "It also activates several other pathways, such as JAK-2/STAT" - JAK2 activation is central to GH receptor binding!, STAT (especially STAT5b) also plays a major role, but as you point out, other signaling pathways such as mTOR and MAPK are important in intracellular GH action
* line 59: instead of writing "of some tumors (breast, colon, [...]" rather use "of tumors such as breast, colon, [...]"
METHODS
The methods are clearly described. However, there are a few aspects not covered:
As some research groups have reported an increased frequency of thyroid carcinoma in patients with acromegaly, it would be important for the authors to clarify their method for differentiating benign from malignant nodules.
The study design could have benefited from the inclusion of a control group. For instance, a group of nonfunctioning pituitary adenoma (NFPA) patients or healthy individuals could have provided a comparative baseline to gauge the observed effects more effectively. This comparison could potentially strengthen the findings by illustrating the relative prevalence of tumors in the study population.
further comments:
line 95/96: "according to the national guidelines for the diagnosis and management of acromegaly" - reference missing
line 97/98: "Information regarding the patient’s benign and malignant tumors were retrieved from the hospital’s database" - systematically screened for lesions (besides thyroid ultrasound)?
line 107/108: "The level of significance was set at p=0.05 for the whole study." - more correct: "p < 0.05"
RESULTS
The results section, as it should, effectively presents the data collected during the study. The data, as presented, are predominantly descriptive in nature, with some comparisons between groups. I believe the authors could delve deeper into the data to provide a more nuanced and insightful analysis. A comparison of risk factors between those with malignant tumors and those without any tumors would have been a valuable addition to the current results. Furthermore, a comparison between malignant and benign tumors may yield significant insights. This would help to differentiate between these two distinct types of tumor and might uncover any specific factors that are more strongly associated with malignant tumors. It would be particularly interesting to know if there are any specific risk factors that are significantly more prevalent in one group over the other. (Of course, acknowledging the limitation of a fairly small group.)
To optimize the impact and relevance of the findings, I suggest that the authors consider these additional analyses. This would not only enhance the comprehensiveness of the results but also provide more depth to the implications of the findings.
further comments:
* table 2: What is the rationale for multiplying IGF-1 ULN with the estimated diagnostic delay? Can you provide a reference?
DISCUSSION
The discussion sections offers a reasonable interpretation of the study's results. However, the quality of the written text could be considerably improved with serious copy editing. There are sections where the prose could be smoother, and the logical flow of ideas could be better organized. Streamlining the text and improving the syntax and punctuation where needed could significantly enhance the readability and impact of the discussion.
In this section, the authors made reference to data from the Global Cancer Observatory, which I found to be a noteworthy and intriguing addition. This comparative context provides an additional dimension to the research findings. Nevertheless, I believe the manuscript could benefit from an extended discussion of these data. Further elaboration on how their findings compare or contrast with this global data set, and potential reasons for any discrepancies, could provide further depth and relevance to the study. Furthermore, it would be beneficial if the authors could also explain what Globocan is, for the benefit of readers who might not be familiar with this resource.
further comments:
* lines 196ff.: "The result of our study found no significant differences between men and women regarding the age at diagnosis, GH and IGF-1 levels, or duration of the disease. However, there was a predominant female population of acromegalic patients." - please rephrase
* lines 247/258: the content is a duplicate of lines 207/208
CONCLUSIONS
The statement beginning in line 325 is not a conclusion from this study, so please remove it.
The manuscript is mostly well written and readable, but the wording is odd at times and imprecise.
Reviewer 2 Report
It is an original article addressing an old topic, namely the tumor risk in acromegalic patients.
The authors found no relationship between the levels of IGF-1 or GH and tumor development, but showed that certain risk factors, such as diabetes mellitus and obesity, were more frequent in acromegalic patients with tumors.
The results are not of great value, being similar to those mentioned in previous studies on larger case series. The article is written rather extensively but the results, although poor, are clearly presented.
The title of the article is explicit and related to the content of the manuscript.
I have to address a few minor comments that the authors should consider.
The abstract is well structured, with relevant content. I would propose changing the conclusion, namely that - a direct link between acromegaly and the presence of tumor formations could not be found, but that certain risk factors are more common in acromegaly with tumors.
Otherwise, in line 31, the word obesity appears instead of acromegaly.
The introduction should generally have a fragment at the end in which the objectives of this study are presented.
Methods. There are some gaps in the design of the study. There is too long a presentation of the diagnosis of acromegaly which is well known and almost nothing about the identification of tumors. It is not specified at what point in the disease the data were collected. At diagnosis, during treatment? What investigations do your national guidelines include? Colonoscopy, thyroid evaluation ... Was the polysomnographic evaluation performed on all patients? Mammography for all women? PSA/prostatic US for all men?
A shortcoming of the study is that there was no prospective assessment of tumor identification, especially of common ones (such as skin, breast), it is not clear whether all patients had colonoscopy/polysomnography and at what rate and if they were treated, with functional improvement. I noticed the incomplete presentation of comorbidities, there is no information about hypertension, cardiovascular status…. It is not shown whether the diabetes improved/cured after surgical/medical treatment.... Do you have information about insulin resistance?
I think that the discussions are too extensive, and the reader gets lost in irrelevant details. Please review this part. Rephrase the conclusions accordingly.
Minor editing of English language required
Reviewer 3 Report
The present study about benign and malignant tumour association with acromegaly is well written and documented, the methods and results sound.
The authors describe and discuss many evidence in the literature although they do not take at all in consideration any possible genetic features favouring, or being related to such disease association (please discuss, add details if necessary).
Another possibility to greatly improve this study could be to perform a literature review and or a metaanalysis of available study but I understand that the initial study design is far from this proposition.
Round 2
Reviewer 1 Report
I appreciate the effort that was made in improving the paper and answering my numerous questions/remarks. I think the manuscript has benefited from this.
As conceded by the authors themselves, a language editing service for the final version of the paper is necessary.